# Consistency between Primary Uterine Corpus Malignancies and Their Corresponding Patient-Derived Xenograft Models

**DOI:** 10.3390/ijms25031486

**Published:** 2024-01-25

**Authors:** Shoko Ueda, Tomohito Tanaka, Kensuke Hirosuna, Shunsuke Miyamoto, Hikaru Murakami, Ruri Nishie, Hiromitsu Tsuchihashi, Akihiko Toji, Natsuko Morita, Sousuke Hashida, Atsushi Daimon, Shinichi Terada, Hiroshi Maruoka, Yuhei Kogata, Kohei Taniguchi, Kazumasa Komura, Masahide Ohmichi

**Affiliations:** 1Department of Obstetrics and Gynecology, Educational Foundation of Osaka Medical and Pharmaceutical University, 2-7 Daigakumachi, Takatsuki 569-8686, Osaka, Japan; shouko.ueda@ompu.ac.jp (S.U.); shunsuke.miyamoto@ompu.ac.jp (S.M.); hikaru.murakami@ompu.ac.jp (H.M.); ruri.nishie@ompu.ac.jp (R.N.); hiromitsu.tsuchihashi@ompu.ac.jp (H.T.); akihiko.touji@ompu.ac.jp (A.T.); natsuko.morita@ompu.ac.jp (N.M.); sosuke.hashida@ompu.ac.jp (S.H.); atsushi.daimon@ompu.ac.jp (A.D.); shinichi.terada@ompu.ac.jp (S.T.); hiroshi.maruoka@ompu.ac.jp (H.M.); yuhei.kogata@ompu.ac.jp (Y.K.); m-ohmichi@ompu.ac.jp (M.O.); 2Center for Medical Research & Development, Division of Translational Research, Osaka Medical and Pharmaceutical University, 2-7 Daigakumachi, Takatsuki 569-8686, Osaka, Japan; kohei.taniguchi@ompu.ac.jp (K.T.); kazumasa.komura@ompu.ac.jp (K.K.); 3Department of Regenerative Science, Graduate School of Medicine, Dentistry and Pharmaceutical Sciences, Okayama University, 2-5-1 Shikatachou, Kita-ku, Okayama 700-8558, Okayama, Japan; pmes391t@s.okayama-u.ac.jp

**Keywords:** uterine corpus malignancies, patient-derived xenograft, heterotopic implantation, extracellular vesicles, personalized medicine, translational research, DNA, RNA

## Abstract

Patient-derived xenograft (PDX) models retain the characteristics of tumors and are useful tools for personalized therapy and translational research. In this study, we aimed to establish PDX models for uterine corpus malignancies (UC-PDX) and analyze their similarities. Tissue fragments obtained from 92 patients with uterine corpus malignancies were transplanted subcutaneously into immunodeficient mice. Histological and immunohistochemical analyses were performed to compare tumors of patients with PDX tumors. DNA and RNA sequencing were performed to validate the genetic profile. Furthermore, the RNA in extracellular vesicles (EVs) extracted from primary and PDX tumors was analyzed. Among the 92 cases, 52 UC-PDX models were established, with a success rate of 56.5%. The success rate depended on tumor histology and staging. The pathological and immunohistochemical features of primary and PDX tumors were similar. DNA sequencing revealed similarities in gene mutations between the primary and PDX tumors. RNA sequencing showed similarities in gene expressions between primary and PDX tumors. Furthermore, the RNA profiles of the EVs obtained from primary and PDX tumors were similar. As UC-PDX retained the pathological and immunohistochemical features and gene profiles of primary tumors, they may provide a platform for developing personalized medicine and translational research.

## 1. Introduction

Uterine corpus cancer is the sixth most commonly diagnosed cancer in women, with approximately 417,000 novel cases and 97,000 deaths in 2020 [1]. The majority of endometrial cancers are detected at an early stage and have favorable outcomes, with a 5-year survival rate of approximately 90% [2,3]. However, patients with high-grade, advanced, or recurrent disease have poor prognosis. Treatment options for these patients remain limited, and our understanding of high-risk diseases has to be improved to develop novel therapies that can fill the treatment gaps.

The establishment of an effective preclinical animal research platform is essential for basic and translational studies on cancer. The use of cancer cell lines has led to significant advancements in cancer biology and is crucial for cancer research [4]. Although cell lines exhibit identical gene profiles and reactions in certain situations, they do not adequately predict drug sensitivity [5,6,7]. To overcome these limitations, patient-derived xenograft (PDX) models are being increasingly used as advanced preclinical cancer models. PDX models are established by directly engrafting fresh patient tumors into immunocompromised mice, maintaining tumor growth in vivo. These models retain the biological and molecular characteristics and heterogeneity of patient tumors better than cell lines [8,9,10,11,12,13,14]. Therefore, they are used for the assessment of anti-tumor drug efficacy, discovery of novel anti-cancer medicines, identification of biomarkers, development of personalized medicine, biological studies, and evaluation of drugs at the preclinical stage [13,15,16,17,18,19,20].

An increasing number of PDX models representing different types of cancers, such as those of the breast [21], lung [22], pancreas [23], colon [24], kidney [25], bladder [26], cervix [27], and ovary [28], as well as melanoma [29], are being developed. However, reports on PDX models of endometrial cancer are limited, and evaluation of how accurately they represent and correlate with a patient’s disease is scarce [10,30].

Recent studies have shown that extracellular vesicles (EVs) are involved in cancer progression and metastasis and that EVs contain proteins and nucleic acids inside and function as mediators of intercellular signaling between cancer cells and the microenvironment [31,32]. Targeting the secretion and cargoes of EVs has become an important new field of cancer therapeutic research [31,32,33]. Therefore, we also investigated EVs in PDX tumors.

Hence, the purpose of this study was to use the PDX models of uterine corpus malignancies (UC-PDX) to determine whether the characteristics identified in the primary tumor (pathological findings, genetic mutations, and gene expression) were consistent with those of the corresponding PDX tumor. Furthermore, the RNA in EVs was analyzed to evaluate the similarity between primary and PDX tumors.

## 2. Results

### 2.1. Establishment of the UC-PDX Models

Among the 92 samples obtained from patients with uterine corpus malignancies, 56 UC-PDX were established, with a transplantation success rate of 56.5% (52/92). The patient characteristics and PDX engraftment rates are shown in Table 1. The engraftment rate did not depend on patient age, carbohydrate antigen 19-9 (CA19-9) and carbohydrate antigen 125 (CA125) levels, lymph node metastasis, or peritoneal cytology positivity. It was not significantly different between the patients with high levels of CA125 or CA19-9 and those with low levels of them. However, it was notably higher in those with high-grade tumors, advanced disease, and lymphovascular invasion. Therefore, we showed that tumors with poor clinical prognosis have a high PDX engraftment rate.

### 2.2. Histological Evaluation of Patient and PDX Tumors

Figure 1a shows the pathological and immunohistochemical features of primary (P) and PDX tumors (F0 and F2) in the one case of grade 1 endometrioid carcinoma (PDX 214) as a representative example. The pathological features of P, F0, and F2 were similar, and the tumors were usually composed of columnar cells with pseudostratified nuclei. They were characterized by a back-to-back glandular arrangement or a crowded, branching, and cribriform architecture. In immunohistochemical analysis, the tumor cells were positive for estrogen receptor alpha (ERα) and negative for progesterone receptor (PR) in all samples, including P, F0, and F2. The p53 index was 0% in all samples. The Ki-67 indices for P, F0, and F2 were 67%, 60%, and 41%, respectively. Figure 1b shows the pathological and immunohistochemical features of P and PDX tumors (F0 and F2) in the one case of serous carcinoma (PDX157) as a representative example. The pathological features of P, F0, and F2 were similar; the tumors exhibited complex papillary and glandular architectural features, with severe pleomorphism, conspicuous nucleoli, and frequent mitoses. In immunohistochemical analysis, the protein expression of ERα and PR were negative in all samples. The p53 and Ki-67 indices were 67% and 62% for P, 46% and 63% for F0, and 59% and 56% for F2, respectively.

### 2.3. Mutations in Primary and PDX Tumors

Amplicon sequencing was performed to identify the genomic mutations in the 26 PDX models. The samples comprised seven endometrioid carcinomas of grade 1, five endometrioid carcinomas of grade 2, five endometrioid carcinomas of grade 3, three serous carcinomas, four carcinosarcomas, and two sarcomas. Figure 2a,b shows the genetic mutations in each sample. In all cases, most genetic mutations in the primary tumors were present in the PDX tumors (F0). In total, 62.5–100% of genetic mutations detected in primary tumors were inherited by the F0 tumors of the 26 PDX models. Mutations inherited from primary to F0 were maintained in successive generations, and the same tendency was observed for F2, F4, and F6. However, in most cases, the F0 models had more mutations than the primary tumors. Conversely, the number of mutations in F2, F4, and F6 tumors was similar to that in F0 tumors; the number of mutations did not increase after the establishment of implantation. In most cases, non-frameshift substitutions of platelet-derived growth factor receptor alpha polypeptide (*PDGFRA*) and frameshift deletions of adenomatous polyposis coli (*APC*) have been found. Similar mutations were observed in the primary and PDX tumors (F0, F2, F4, and F6). In most cases, mutations in *NRAS*, serine/threonine kinase 11 (*STK11*), and isocitrate dehydrogenase (*IDH2*) were detected in PDX but not in primary tumors. Figure 2c,d shows the correlations between the variant allele frequencies (VAFs) of somatic mutations identified in the primary and PDX (F0 and F2) tumors. The correlation coefficient of the VAFs between the primary tumor and F0 was 0.235–0.996. The correlation coefficient was higher than 0.7 in 96% of cases (25/26), showing a strong positive correlation in most cases. The correlation coefficient of the VAFs between the primary tumor and F2 was 0.839–0.991; the concordance rates between the primary and PDX tumors (F0 and F2) were high.

### 2.4. Transcription Profile of Primary and PDX Tumors

RNA sequencing was performed using 22 PDX models for genomic expression analysis. The samples comprised seven endometrioid carcinomas of grade 1, four endometrioid carcinomas of grade 2, two endometrioid carcinomas of grade 3, three serous carcinomas, four carcinosarcomas, and two sarcomas. Figure 3a,b show the gene expression in the primary and PDX tumors using hierarchical heat map clustering analysis. In most cases, *GNAS* and nucleophosmin (*NPM1*) were expressed at high levels, whereas FMS-like tyrosine kinase 3 (*FLT3*) was expressed at low levels. The genomic expression profiles of the primary (P) and PDX tumors (F0, F2, F4, and F6) were similar. The genomic expression levels of the colony-stimulating factor 1 receptor (CSF1R), kinase insert domain receptor (KDR), Janus kinase 3 (JAK3), and PDGFRA were lower in PDX than in primary tumors. Figure 3c,d shows the pair correlation plots of genomic expression in primary and PDX (F0 and F2) tumors. The correlation coefficient of genomic expression between the primary tumor and F0 was 0.557–0.946. The correlation coefficient was higher than 0.7 in 82% of cases (18/22). A strong correlation was observed between the primary and PDX tumors (F0). The correlation coefficients between the primary and F2 tumors were between 0.622 and 0.891. The correlation between gene expression in the primary and PDX tumors did not decrease over successive generations.

### 2.5. RNA Expression of Tissue-Exudative Extracellular Vesicles in Primary and PDX Tumors

The presence of tissue-exudative extracellular vesicles (Te-Evs) was confirmed using Western blotting (Figure 4a), nanoparticle tracking analysis (NTA) (Figure 4b), and electron microscopy (Figure 4c).

RNA sequencing was performed to determine the genomic expression of Te-Evs in six PDX models. The samples consisted of two endometrioid carcinomas of grade 1, three endometrioid carcinomas of grade 3, and one carcinosarcoma. Figure 5a shows the gene expression in the Te-Evs of primary and PDX tumors using hierarchical heat map clustering analysis. As observed previously, *GNAS* and *NPM1* were expressed at high levels, whereas *FLT3*, rearranged during transfection (*RET*), and anaplastic lymphoma kinase (*ALK*) were expressed at low levels. Figure 5b,c shows pair correlation plots of genomic expression in the Te-Evs of primary and PDX (F0 and F2) tumors. The correlation coefficient of genomic expression between the primary tumor and F0 was 0.558–0.956. The correlation coefficient was higher than 0.7 in 83% of cases (5/6). The genomic profiles of Te-Evs from the primary and PDX tumors (F0) were similar.

## 3. Discussion

UC-PDXs were established in 52 patients, with an engraftment rate of 56.5% (52/92). The pathological and immunohistochemical features of the primary tumor and UC-PDXs were similar. Genomic features, including mutations and gene expression, of the primary tumors and UC-PDXs were similar. Genomic expression in EVs from primary tumors and UC-PDXs was also similar.

### 3.1. Success Rate and Transplantation Method

In general, engraftment rates are influenced by multiple factors, including the stage and histology of the primary tumor, site of transplantation, preservation of the tumor specimen, and type of immunocompromised mouse [8]. In previous studies, the engraftment rate depended on the tumor grade in endometrioid carcinomas [18,19]. In a study by Bonazzi et al., histological grade 1 tumors were not engrafted, but grade 2 and 3 tumors were successfully engrafted, and the rate for successful engraftment of grade 3 tumors was higher than that for grade 2 tumors (33 vs. 61%) [18]. In our study, the success rate depended on tumor histology and staging. Additionally, the engraftment rate was significantly higher in tumors with lymphovascular invasion. In lung cancer, advanced-stage tumors and poorly differentiated cancers have higher successful engraftment rates [34].

Transplantation sites are of two types: heterotopic and orthotopic. Subcutaneous transplantation is the most common mode of heterotopic implantation, while others include implantation in the subrenal capsular and interscapular regions [10]. In UC-PDX models, orthotopic implantation involves direct implantation into the uterus via transvaginal injection of a minced tumor or surgical implantation into the uterine cavity [35,36]. Although subcutaneous PDX is relatively simple to create and tumor growth can be monitored visually and palpated, metastasis is rare, and complete simulation of the microenvironment of the primary tumor is difficult [30]. Conversely, orthotopic implantation can mimic the environment of the primary tumor, and two reports of metastases are available [35,36]. However, orthotopic PDX has two disadvantages: the procedure is complicated and imaging techniques are required to determine tumor growth, as it cannot be determined by surface observation [36]. In our study, subcutaneous implantation was performed in all cases; however, none of the patients developed metastases.

Ideally, the establishment of a PDX model would involve the storing of patient material and the growing of PDX material to re-implant when required because of limited funding and the difficulty in obtaining tissue specimens. However, in practice, fresh tissue is almost always used, and Bonazzi et al. reported that cryopreserved uterine cancer tissue has a lower transplantation success rate than fresh tissue [18]. Cryopreservation methods using various concentrations and combinations of fetal calf serum (FCS), dimethyl sulfoxide (DMSO), and ethylene glycol have been reported for several cancers [37,38,39,40]. Alkema et al. showed that collection rates of 38% and 67% were achieved using vitrification or FCS/DMSO-based cryopreservation of primary tumors, respectively, compared to 61% obtained using fresh PDX tumors. Similar results were obtained for the established PDX tumors, indicating a higher success rate with FCS/DMSO-based cryopreservation than with vitrification [38]. Cui et al. reported transplantation failure in all cases of cryopreserved primary colorectal and pancreatic cancer tissues, compared to a 63% transplantation success rate of fresh tissue. However, a success rate of 39% was achieved only after cryopreservation of tumors that had already been xenografted [39]. In established colorectal carcinoma PDX models, Linnebacher et al. did not find any difference between the cryopreserved cancer tissues and fresh cancer samples (74% vs. 71%). Furthermore, re-transplantation of xenograft tumors after cryopreservation was successful for 11 tumors [40]. To the best of our knowledge, a reliable and reproducible cryopreservation protocol for UC-PDX has not yet been reported. However, primary transplantation of fresh patient tumors into mice, followed by cryopreservation of well-established PDX tissues, is the preferred method for efficient PDX biobanking.

The use of immunocompromised mice is essential for the establishment of PDX models, and higher levels of immunodeficiency are usually associated with more successful PDX engraftment [24]. However, nude mice are often used for gastrointestinal tumors because they have a relatively high engraftment rate, can be used for subcutaneous tumor observation, and are inexpensive [24,41,42]. In gastric cancer, patients with advanced disease have higher IgG levels than those with early-stage disease, and IgG is expected to play an important role in tumor proliferation and infiltration. SCID mice lose existing B and T cells. Contrarily, although the nude mice lost their B cell function, the number of B cells remained normal. Thus, nude mice are more suitable as PDX models for gastric cancer [43,44]. Although the engraftment rate of UC-PDX was 56.5% in NOD/SCID mice in this study, Shin et al. reported that the success rate of endometrial cancer PDX was 56.3% (18/32) in nude mice and suggested that PDX models of gynecological cancer did not correspond to the immunodeficient grades [19]. The contamination of PDX models with lymphoproliferative lesions is one of the challenges in using immunocompromised mice. Most studies have reported the development of Epstein–Barr virus (EBV)-related B-cell lymphoma due to inadequate immune surveillance in mice [45,46]. Choi et al. showed that EBV-related B-cell lymphomas developed in NOG (NOD/Scid/IL2Rγnull) mice but not in nude mice. This may be due to the action of natural killer cells in the nude mice [47]. Therefore, establishing UC-PDX using nude mice may reduce costs and prevent lymphoma development.

### 3.2. Comparison of Original Tumors and PDXs

Most studies have confirmed high similarities in the histology, phenotypic features, and molecular markers of UC-PDX models and primary tumors [9,11,12,13,18,30,35,36,48]. The characteristics identified in the primary tumor were retained after repeated passages [9,11,18]. Using immunohistochemistry, several studies have shown that UC-PDX tumors express biomarkers similar to those of primary tumors, such as estrogen receptors, progesterone receptors, p53, and Ki-67 [9,35,48]. Similar results were observed in the present study.

Recently, several studies have reported the genetic analysis of UC-PDX. Depreeuw et al. performed whole-exome sequencing (WES) on four models (two endometrioid, one mesonephric, and one serous carcinoma). The majority of the mutations were common between primary tumors and F3 PDX tumors (55%); analyses specifically of the cancer consensus genes from COSMIC revealed that most of the mutations were common. On average, 90% of the genome of primary and F3 PDX tumors had the same copy number [9]. Zu et al. performed WES and RNA sequencing to compare primary and F4 PDX tumors in two high-risk endometrial carcinomas. The DNA mutation frequencies of primary and F4 PDX tumors showed significant linear correlation, and their gene expression profiles were similar [12]. Bonazzi et al. performed WES on endometrial cancers with four common molecular subtypes: those with DNA polymerase epsilon (POLE) mutations, mismatch repair deficiency (MMRd), p53 mutations, and without any specific molecular profile. They focused on the MMRd subtype because it was expected to accumulate changes during passages because of a lack of DNA mismatch repair. They reported that mutational heterogeneity was more frequent in the MMRd models than in the non-MMRd models. In the p53 mutation subtype, the total number of somatic mutations was consistent between primary and PDX tumors [18]. Villafranca-Magdalena et al. performed WES of 13 endometrial carcinoma (seven endometrioid and six serous carcinomas). Based on The Cancer Genome Atlas (TCGA) classification, all endometrioid carcinomas were classified as the microsatellite instability (MSI) type, and all serous carcinomas were classified as the high copy number (HCN) type. They compared all genes carrying single-nucleotide variants (SNVs) and copy number variations (CNVs) between PDX and primary tumors and found that the SNVs, but not CNVs, of MSI PDX were highly similar to those of the primary tumor. Conversely, HCN PDX reliably reproduced the CNV profile, although the SNVs were not similarly conserved [13]. By performing DNA and RNA sequencing of 50 oncogenes and tumor suppressor genes, our study showed that most genomic features of the original uterine corpus malignancies were retained in PDX models throughout the xenografting and passaging processes.

In addition, we performed RNA sequencing of the Te-EVs extracted from primary and PDX tumors. EVs are composed of a lipid bilayer with transmembrane proteins and contain cellular proteins, lipids, and nucleic acids (microRNAs, mRNA, and DNA) [49]. Cancer-associated EVs play critical roles in establishing the tumor microenvironment, tumor growth, and metastasis. Although cell culture-derived EVs and body fluid-derived EVs have been used, Te-EVs have only recently attracted attention [50,51]. Compared to body fluid-derived EVs, Te-EVs can minimize contaminants and accurately reflect the pathophysiological characteristics and behavior of cells, as the three-dimensional structure of the tissue and the properties of the cells are retained [52]. However, reports on EVs extracted from PDXs are limited [53]. In this study, we collected and compared EVs from primary and PDX tumor tissues. We have previously reported that mRNAs in the Te-EVs derived from primary and PDX tumors of cervical cancer were similar [14], and the same results were obtained in uterine corpus malignancies. In summary, high recapitulation of the histological characteristics, gene mutations, and gene expression profiles was observed in our study when UC-PDX models were compared with the primary tumor.

Here, all primary tumors and xenografts harbored genetic mutations in *PDGFRA* and *APC*. PDGFRA belongs to the type III receptor tyrosine kinase family and modulates nuclear regulatory proteins [54]. APC encodes a large multidomain protein that plays an integral role in the Wnt-signaling pathway and intercellular adhesion [55]. Contrarily, genetic mutations in *NRAS, STK11*, and *IDH2* were found only in xenografts and were absent in primary tumors. We also observed that the PDX tumors harbored a higher number of mutations than the primary tumors. *NRAS* mutations constitutively activate intracellular signaling via various pathways, most notably the RAS-RAF-MAPK and PI3K-AKT pathways. These activated signaling pathways induce cell cycle dysregulation, pro-survival pathways, and cellular proliferation [56]. STK11 functions as a tumor suppressor and is involved in several pathways that control cell growth and apoptosis [57]. IDH is a major enzyme in the citric acid cycle and is believed to function in maintaining the redox state of cells, playing an important role in cellular defense against oxidative stress [58]. Although these PDX-specific mutations may be the result of adaptation to transplantation in a novel microenvironment, some studies have reported that they are the results of enrichment of genomic alterations in the primary tumor of PDX tumors [59,60]. In addition, PDX tumors are possibly affected by the loss of human stromal cells and their replacement by the stroma of mouse origin during growth. [18,61]. However, the mutations inherited by the PDX (F0) tumor from the primary tumor were maintained in the mouse-to-mouse passages, and the number of mutations rarely increased once the transplant was established, suggesting that the PDX genome was relatively stable.

### 3.3. Clinical Applications of the PDX Model

There have been several studies of PDX as an extraordinary preclinical tool for testing drug sensitivity and the efficacy of approaches in personalized medicine. Berotti et al. found that amplification of the Erb-B2 receptor tyrosine kinase 2 (*ERBB2*) gene is a determinant of the driver of cetuximab resistance and predicts the response to epidermal growth factor receptor (EGFR) and HER-2-targeted therapies in a colon cancer PDX model [62]. The results described above were derived from clinical trials [63,64,65]. Other biomarkers for cetuximab resistance were also identified in colon cancer PDX [66,67,68,69,70]. An “avatar” is defined as a PDX that received the same anticancer agent that the donor patient received. In colon cancer, an avatar trial is ongoing [71]. Stebbing et al. performed the first study using PDX mice to evaluate the therapeutic response of individual patient-derived tumors. Several treatment options, including chemotherapeutic agents, tyrosine kinase inhibitors, and combination regimens, were evaluated and ultimately applied in individual PDX for patients with advanced metastatic sarcoma, with a correlation of tumor response rate in 13 of 16 (81%) patients [72]. In a study by Hidalgo et al., surgically excised tumor grafts from 14 patients with refractory advanced cancers were generated and tested against 63 drugs. A total of 12 patients received the therapy that worked best in the PDX mice and achieved partial remissions, with an overall strong correlation between PDX and patient outcomes [73]. Sorokin et al. evaluated combined MEK-CDK4/6 inhibition in RAS mutant colorectal cancer in the co-clinical trial using PDX. They demonstrated therapeutic efficacy in PDX and safety in patients, identified biomarkers of response, and uncovered targetable mechanisms of resistance [74]. Several large PDX clinical trials have examined drug response and resistance mechanisms, and the results confirm the reproducibility and clinical applicability of PDX [75,76,77]. This indicates that PDX retains therapeutic accuracy and is a clinically functional model.

### 3.4. Problems in the PDX Model

PDX tumors established in immunocompromised mice cannot reproduce the human immune microenvironment. This makes it difficult to apply them to the exploration of issues related to immunotherapy, such as immune checkpoint inhibitors. Therefore, humanized mice have recently attracted attention as an effective tool for immunotherapy evaluation [78]. Humanized mice are created by irradiating immunocompromised mice and then transplanting human CD34+ hematopoietic cells, human peripheral blood mononuclear cells, or bone marrow–liver–thymus to reproduce the human immune system [79,80,81]. Humanized mice implanted with tumor fragments derived from patients are called humanized PDX [82]. However, humanized PDX still needs to resolve several technical issues before it can be truly effective for use in nonclinical trials due to its cost, short lifespan, unstable engraftment, and onset of graft-versus-host disease [81,83]. In addition, there are still many issues such as major histocompatibility complex incompatibility between immune cells and tumors, residual mouse innate immune cells, and lack of specific cytokines, which may cause the therapeutic response in humanized mice to be different from that of patients [81].

Comparison of primary and PDX tumors has shown high fidelity, but there are some limitations. Cancer cells are embedded in a supportive tumor microenvironment. This microenvironment includes stromal cells, which are composed of tumor endothelial cells and cancer-associated fibroblasts, as well as tumor-associated macrophages that have a significant effect on cancer progression and metastasis [84]. However human stromal components are rapidly lost during the process of engraftment and replaced by the mouse microenvironment [85]. The exact impact of murine stroma on human cancer cells in PDX models remains unknown. Moreover, due to intratumoral heterogeneity and limited tissue quantities, PDX tumors can hardly represent the whole landscape of primary tumors [86]. In a study of one patient with high-risk neuroblastoma, a total of 10 PDX tumors were established. The growth rates of these xenograft tumors varied significantly in the same generation. Additionally, RNA-seq and proteomic analyses have demonstrated transcriptional and translational variations among these PDX tumors [87]. Therefore, PDX models must undergo selection bias. Furthermore, once primary tumors were injected into mice, only a fraction of cancer cells were competent to engraft, which reduces genetic diversity owing to out-competition by the fittest and most rapidly proliferating clone. In short, xenograft tumors were exposed to mouse-specific selection pressure [20].

This study has several limitations. First, DNA and RNA sequencing can only be used to analyze specific cancer-related genes and other major genes were not included. Second, although several mutations have been identified as potential therapeutic targets, drug susceptibility tests have not been performed. Further studies, including those on WES and drug sensitivity, are required.

## 4. Materials and Methods

### 4.1. Patients and Tissue Samples

Tumor samples were obtained from 92 patients diagnosed with uterine corpus malignancy who underwent laparoscopic, robot-assisted, or abdominal hysterectomy at the Osaka Medical and Pharmaceutical University in Japan between April 2019 and April 2022. None of the patients received any therapy, including chemotherapy or radiotherapy, before surgery. Written informed consent was obtained from all patients. This study was ethically reviewed and approved by the Ethics Committee of Osaka Medical and Pharmaceutical University (Assurance Number 2191).

Fresh tumor tissues were rinsed, cleared of blood and necrotic tissue, and cut into four sections. The first section was immediately placed in Dulbecco’s modified Eagle’s medium nutrient mix F-12 (DMEM/F12, Gibco, Thermo Fisher Scientific, Tokyo, Japan), minced sharply, and mixed with Matrigel (Corning, New York, NY, USA) on ice for transplantation. The second section was fixed in 10% formalin and embedded in paraffin (FFPE) for pathological analysis. The third section was placed in RNAlater tissue storage reagent (Thermo Fisher Scientific) and stored at −20 °C for gene analysis. The fourth section was immersed in a serum-free medium (FujiFilm Wako Pure Chemical Corporation, Osaka, Japan) with exosome-depleted FBS (System Biosciences, Palo Alto, CA, USA) for purification of Te-EVs.

### 4.2. Animals

In this study, 4-to-8-week-old female NOD/SCID mice (NOD.CB17-Prkdc^scid^/J, Oriental BioService, Kyoto, Japan) were used for the implantation of uterine corpus malignant tissue. The NOD/SCID mice were housed in a specific-pathogen-free environment at 24–26 °C with 30–50% humidity and had free access to sterile water and standard rodent chow. The mice were euthanized via cervical dislocation by trained staff. All animal experiments were approved by the Osaka Medical and Pharmaceutical University Pharmaceutical Ethics Committee (Assurance Number 21007-A).

### 4.3. Establishment of PDX

For developing the PDX model, tumor tissue (3 mm^3^) was minced, mixed with Matrigel (Corning), and injected into the subcutaneous tissue in the dorsal region of NOD/SCID mice using an 18-gauge needle under isoflurane gas anesthesia. After tumor engraftment, the mice were monitored weekly for a maximum period of six months to confirm the successful establishment of the PDX models. Engraftment failure was considered if the tumor nodules were not palpable within six months. The xenograft mice were euthanized, and the tumors were excised when the tumor volume was approximately 1000 mm^3^. A portion of the xenograft tumor was continuously transplanted into NOD/SCID mice. The remaining tumor was fixed in formalin and placed in RNAlater tissue storage reagent and stored at −20 °C. This process was repeated to produce the next generation, and the xenograft tissues were sequentially labeled as F0, F1, F2, and so on. The primary patient tumors were labeled P. The fragment resected from the primary patient’s tumor was implanted into one mouse. We also used one mouse for implantation from the PDX tumor to the next generated PDX mouse; we did not repeat the implantation even if the tumor growth was not observed.

### 4.4. Pathological Analysis Using Immunohistochemistry

FFPE tissue sections from the patient and xenograft tumors were stained with H&E. Immunohistochemistry was performed on deparaffinized FFPE tissue sections using antibodies against ERα (ab 108398, 1:100 dilution; Abcam, Cambridge, MA, USA), PR (#8757, 1:500 dilution; Cell Signaling, Boston, MA, USA), p53 (#2527, 1:100 dilution; Cell Signaling), and Ki-67 (#9027S, 1:200 dilution; Cell Signaling). After incubation with primary antibodies, the slides were rinsed with phosphate-buffered saline (PBS), incubated with a species-specific secondary antibody, and stained using 3,3′-diaminobenzidine substrate solution (MBL, Osaka, Japan). Images were captured using a microscope (BZ-X700 Series, Keyence, Tokyo, Japan). A scale of 0–3 was used for ERα and PR (0: no staining; 1: <5% of the cells showed staining or diffuse staining that was so weak that it could not be distinguished from the background; 2: 6–50% of the cells were stained; 3: >50% of the cells were stained). Staining patterns were grouped as negative (score of 0–1) or positive (score of 2–3) [88,89]. The Ki-67 index was calculated by measuring the mean percentage of Ki-67-positive nuclei among all cells in four different microscopic fields (×200) [90]

### 4.5. Purification of Te-EVs

The tissue samples were immediately immersed in a 9 mL serum-free medium (FujiFilm) with 1 mL exosome-depleted FBS (System Biosciences) and stored at 37 °C for 3 h. The tissue-immersed medium was centrifuged at 2000× *g* for 30 min and filtered through a 0.22 μm filter (Merck Millipore, Bedford, MA, USA) to remove the cell debris. To pellet the EVs, ultracentrifugation was performed at 100,000× *g* at 4 °C for 70 min using an Optima XE-100 (Beckman Coulter, Brea, CA, USA), SW41 T1 (Beckman Coulter), and Ultra-Clear tube (Beckman Coulter). The Te-EVs were collected in PBS and stored at −80 °C (Figure 6).

### 4.6. Western Blot Analysis

The Te-EV samples were dissolved in a radioimmunoprecipitation assay buffer (Thermo Fisher Scientific). Protein samples were separated using sodium dodecyl sulfate-polyacrylamide gel electrophoresis and transferred onto polyvinylidene difluoride membranes. The membranes were blocked using 10% bovine serum albumin in 1× Tris-buffered saline and incubated overnight with specific primary antibodies against CD9 (ab223052, 1:1000 dilution; Abcam), CD63 (sc-5275, 1:200 dilution; Santa Cruz Biotechnology, Dallas, TX, USA), and CD81 (sc-166029, 1:750 dilution; Santa Cruz Biotechnology) at 4 °C. After washing, the membranes were incubated with rabbit immunoglobulin secondary antibody for 1 h. The bands were visualized using an enhanced chemiluminescence reagent (ECL Plus; GE Healthcare Life Sciences, Pittsburgh, PA, USA).

### 4.7. NTA

NTA measurements were performed using a NanoSight NS 300 instrument (Malvern Panalytical, Kanagawa, Japan). The collected EV samples were diluted 1:1000 using PBS before analysis. The samples were loaded into the instrument and analyzed according to the manufacturer’s instructions using the NTA software, version 3.4 (Malvern Panalytical).

### 4.8. Scanning Electron Microscopy

Te-EVs samples suspended in PBS were fixed in 1.25% glutaraldehyde (pH 7.4) and placed on 3 μm polyethylene beads (White Estapor^®^ Microspheres; Merck Chimie S.A.S., Lyon, France) coated with poly-L-lysine. The specimens were then rinsed in PBS, fixed in 1% osmium tetroxide (pH 7.4) for 30 min, and dehydrated using a graded ethanol series. After dehydration, the samples were coated with platinum–palladium and observed using scanning electron microscopy (S-5000; Hitachi, Tokyo, Japan).

### 4.9. DNA and RNA Extraction

DNA and total RNA were extracted from patients and PDX tumors, according to the manufacturer’s protocol, using MagMAX DNA Multi-Sample Ultra 2.0 and MagMAX mirVana total RNA isolation kits (Thermo Fisher Scientific). Total RNA was extracted from Te-EVs using the miRNeasy micro kit (Qiagen, Hilden, Germany). The quantity and quality of the nucleic acids were assessed before library construction using the Qubit dsDNA and RNA high-sensitivity assay kits (Thermo Fisher Scientific), Qubit Fluorometer (Thermo Fisher Scientific), and NanoDrop (Thermo Fisher Scientific).

### 4.10. Amplicon Sequencing

The library was prepared using the Ion Chef System and Ion AmpliSeq Cancer hotspot panel v2 (Thermo Fisher Scientific). This panel was composed of 207 amplicons covering the hotspot regions of 2855 COSMIC mutations in 50 oncogenes and tumor suppressor genes. Sequencing was performed on the Ion GeneStudio S5 Prime using the Ion 540 chip and Ion 540 kit-chef kit (Thermo Fisher Scientific).

### 4.11. RNA Sequencing

Total RNA was reverse transcribed using the SuperScript IV VILO master mix (Thermo Fisher Scientific), followed by library generation using the Ion Chef system, Ion AmpliSeq kit for Chef DL8, and the Ion AmpliSeq RNA cancer panel (Thermo Fisher Scientific). This panel comprises a single pool of primers targeting the same 50 genes addressed in the Ion AmpliSeq cancer hotspot panel v2. Sequencing was performed on the Ion GeneStudio S5 Prime using the Ion 540 chip and Ion 540 kit-chef kit (Thermo Fisher Scientific).

### 4.12. Bioinformatics Analysis

For amplicon sequencing, alignment of the sequences to the reference genome, base calling, and variant calling were performed using the Torrent Suite software Version 5.18.0 (Thermo Fisher Scientific). The mutation annotation format was constructed using the ANNOVAR:20210202 [91]. Somatic mutations with VAFs < 10% were truncated, and genetic mutation analysis was performed using the filtered mutation data.

For the RNA sequencing analysis, the read counts of each mapped gene were normalized to the total mapped fragment counts in each sample and expressed as reads per million mapped reads. This normalization allowed us to compare the expression levels of the target genes between different samples.

All sequencing analyses were performed using supercomputing resources provided by SHIROKANE, the Human Genome Center, Institute of Medical Science, University of Tokyo. Sequencing data were processed using a script written in the programming language, Python (version 3.9.13). Paplot (version 0.5.5) was used as a drawing tool.

### 4.13. Statistical Analysis

Statistical analyses were performed using JMP Pro version 16.2.0 (SAS Institute Japan, Tokyo, Japan). Continuous variables were expressed as mean ± standard deviation. Continuous variables with normal distributions were analyzed using Student’s *t*-test or one-way analysis of variance, and continuous variables with non-normal distributions were analyzed using Wilcoxon tests to assess the differences. The distribution of each factor was assessed using a contingency table and Fisher’s exact test. The chi-square test was used to evaluate the differences between three or more groups. Pearson’s correlation coefficient (R) was used to determine the concordance rate between each pair. Statistical significance was set at *p* < 0.05.

## 5. Conclusions

In this study, we established a UC-PDX model with a viability rate of 56%. PDX tumors retain the characteristics identified in the primary tumor (pathological findings, genetic mutations, and gene expression) to a large extent. The gene expression in Te-EVs also showed a high correlation between the primary and PDX tumors. Therefore, we suggest that the UC-PDX model may be useful as a preclinical model for personalized medicine.

## Figures and Tables

**Figure 1 ijms-25-01486-f001:**
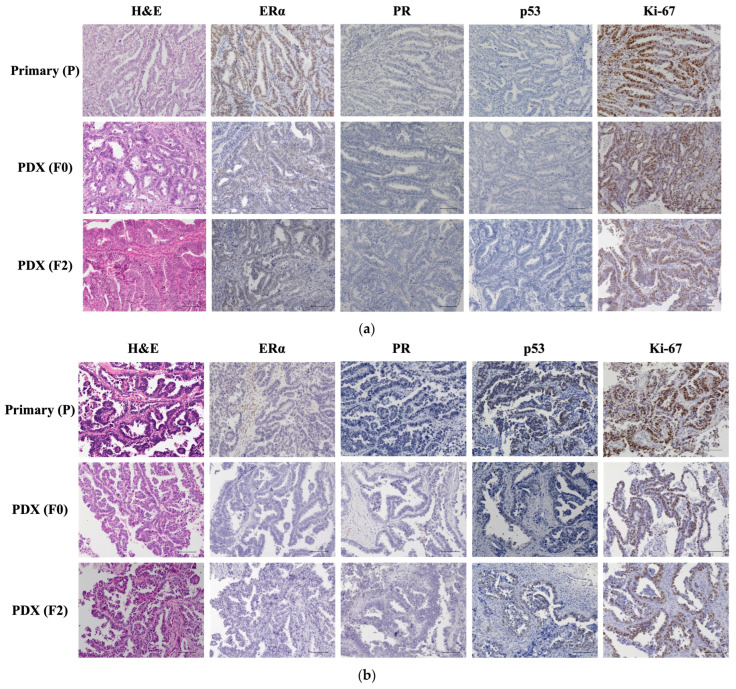
Histopathological comparison of patient primary tumors (P) and the corresponding xenografts (F0 and F2). Representative images of hematoxylin and eosin (H&E) staining and immunohistochemical staining for estrogen receptor α (ERα), progesterone receptor (PR), p53, and Ki-67. The histological features of the xenografts matched well with those of the corresponding patient tumors. Micrographs were captured under 200-fold magnification. Scale bar = 100 μm. (**a**) Staining of primary and PDX tumors in the one case of endometrioid carcinoma of grade 1 (PDX 214). (**b**) Staining of P and PDX tumors in the one case of serous carcinoma (PDX 157).

**Figure 2 ijms-25-01486-f002:**
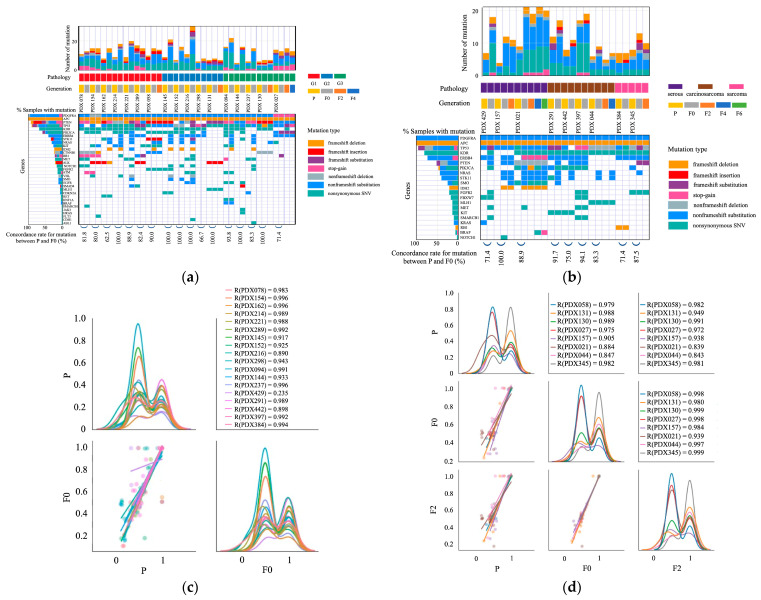
Summary of the gene mutation analysis between primary (P) and PDX tumors (F) based on the results of amplicon sequencing. (**a**) Report of the mutation status in 17 PDX models. The samples consisted of seven endometrioid carcinomas of grade 1, five endometrioid carcinomas of grade 2, and five endometrioid carcinomas of grade 3. The mutation concordance between P and F0 was high in most cases. (**b**) Report on the mutation status in nine PDX models. The samples consisted of three serous carcinomas, four carcinosarcomas, and two sarcomas. In all cases, the mutation concordance between P and F0 was high. (**c**) Variant allele frequencies (VAFs) of somatic mutations identified in P and F0 tumors in 18 PDX models. The lower-left graph shows a scatter plot and linear regression of the VAF levels in the P and F0 tumors. The diagonal graph shows the kernel density estimation (KDE). In most cases, P and F0 tumors showed a high correlation, whereas the PDX429 tumor showed a low correlation. (**d**) VAFs of somatic mutations identified in P, F0, and F2 tumors in eight PDX models. The graph in the middle of the left column shows a scatter plot and linear regression of the VAF levels in the P and F0 tumors. The graph at the bottom of the center column shows a scatter plot and linear regression of the VAF levels in the F0 and F2 tumors. The lower-left graph shows a scatter plot and linear regression of the VAF levels in the P and F2 tumors. The diagonal graph shows the KDE. PDX tumors correlated highly with primary tumors.

**Figure 3 ijms-25-01486-f003:**
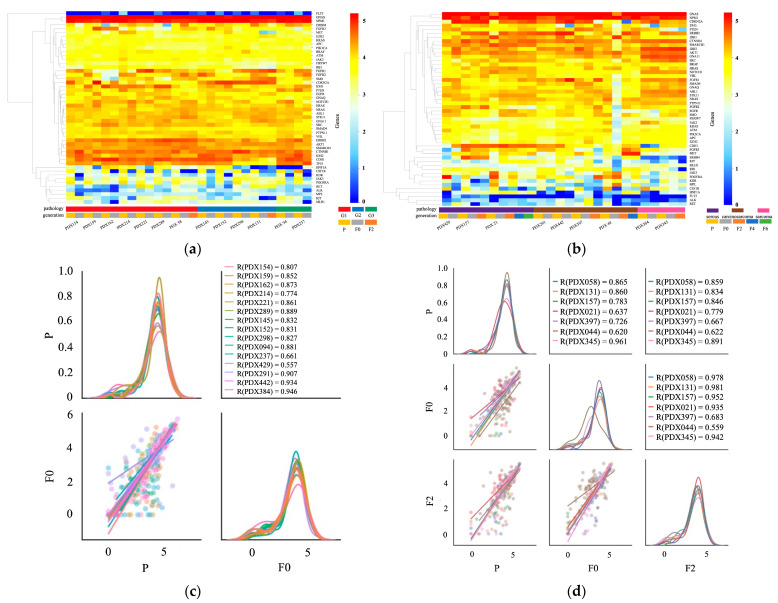
Summary of the gene expression analysis between primary tumors (P) and PDX tumors (F0 and F2) based on the results of RNA sequencing. (**a**,**b**) Hierarchical heat map clustering analysis for gene expression in P and PDX tumors in 21 PDX models. The samples consisted of seven endometrioid carcinoma of grade 1, four endometrioid carcinoma of grade 2, two endometrioid carcinoma of grade 3, three serous carcinoma, four carcinosarcoma, and two sarcomas. In most cases, the gene expression patterns of the primary and PDX tumors were similar. (**c**) Pair correlation plot of the gene expression values between P and F0 tumors in 15 PDX models. The lower-left graph shows a scatter plot and linear regression of the gene expression levels in the P and F0 tumors. The diagonal graph shows the KDE. In most cases, the gene expression in P and F0 tumors showed a high correlation. (**d**) Pair correlation plot of the gene expression values between P, F0, and F2 tumors in seven PDX models. The graph in the middle of the left column shows a scatter plot and linear regression of the gene expression in the P and F0 tumors. The graph at the bottom of the center column shows a scatter plot and linear regression of the gene expression in the F0 and F2 tumors. The lower-left graph shows a scatter plot and linear regression of the gene expression in the P and F2 tumors. The diagonal graph shows the KDE. In most cases, the gene expression of the P and PDX tumors showed a strong positive correlation. Normalized data were converted to base 10 logarithms and z-scores.

**Figure 4 ijms-25-01486-f004:**
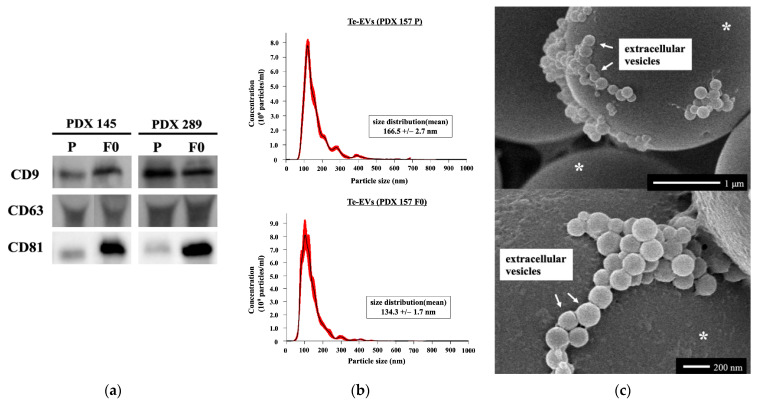
Confirmatory analysis of Te-EVs. (**a**) Western blot analyses were performed to detect the exosomal marker proteins (CD8, CD63, and CD81) in vesicles released by primary and PDX tumors (PDX 145 and 289). (**b**) The nanoparticle tracking analysis was performed to determine the particle size distribution and concentration of Te-Evs extracted from PDX157 prime and PDX tumors. The graphs show the average of five measurements. (**c**) Representative images of Te-Evs obtained using scanning electron microscopy. The visible Te-Evs are approximately 50–200 nm in diameter. White asterisks: 3 μm polyethylene beads.

**Figure 5 ijms-25-01486-f005:**
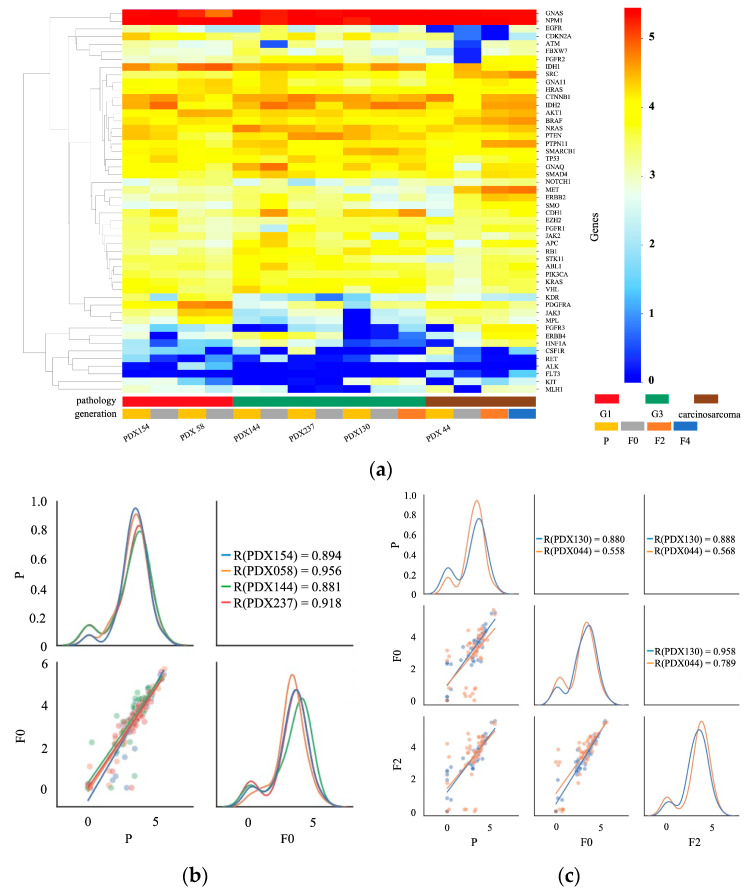
Summary of the gene expression analysis of the tissue-exudative extracellular vesicles (Te-EVs) from primary tumors (P) and PDX tumors (F0 and F2) obtained using RNA sequencing. (**a**) Hierarchical heat map clustering analysis for the gene expression of Te-EVs from six PDX models. The samples consisted of two endometrioid carcinoma of grade 1, three endometrioid carcinoma of grade 3, and one carcinosarcoma. In all cases, the gene expression of the primary and PDX tumors was similar. (**b**) Pair correlation plot of the gene expression values of Te-EVs between P and F0 tumors in four PDX models. The lower-left graph shows a scatter plot and linear regression of the gene expression levels in the P and F0 tumors. The diagonal graph shows the KDE. In all cases, the gene expression in P and F0 tumors showed a high correlation. (**c**) Pair correlation plot of the gene expression values of Te-EVs between P, F0, and F2 tumors in two PDX models. The graph in the middle of the left column shows a scatter plot and linear regression of the gene expression in the P and F0 tumors. The graph at the bottom of the center column shows a scatter plot and linear regression of the gene expression in the F0 and F2 tumors. The lower-left graph shows a scatter plot and linear regression of the gene expression in the P and F2 tumors. The diagonal graph shows the KDE. The gene expression of Te-EVs from P and PDX tumors showed a high correlation. Normalized data were converted to base 10 logarithms and z-scores.

**Figure 6 ijms-25-01486-f006:**
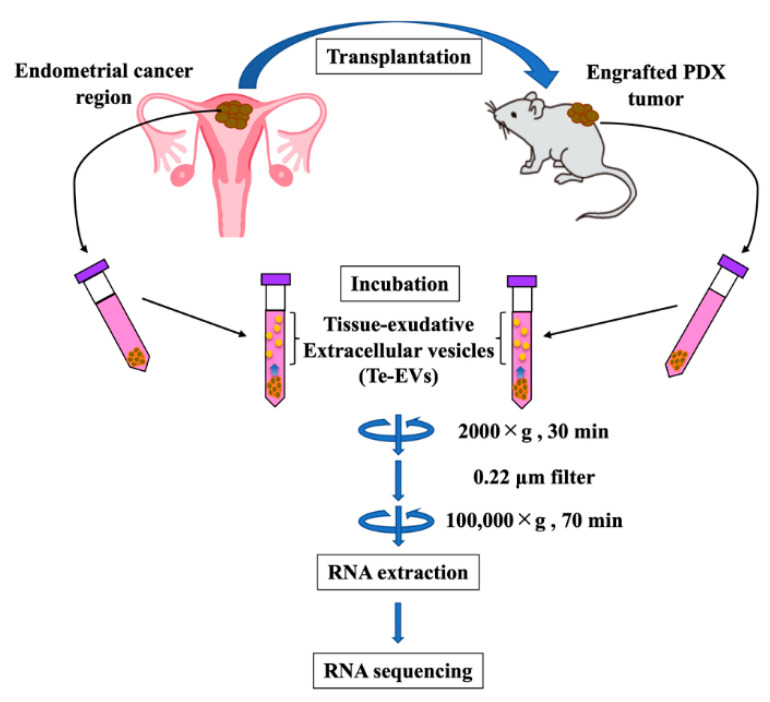
Protocols for the isolation of Te-EVs from fresh surgical tissue and PDX tumors. Total RNA was extracted from the prepared Te-EVs and sequenced.

**Table 1 ijms-25-01486-t001:** Patient characteristics and patient-derived xenograft (PDX) engraftment rate.

Characteristic	Total	Engrafted	Failed	Engraftment Rate (%)	*p* Value
Number of patients	92	52	40	56.5	
Age (years) *	60.9 ± 10.8	61.2 ± 9.4	60.6 ± 12.5		0.782
CA19-9 (U/mL) *	53.0 ± 152.4	59.4 ± 188.4	44.6 ± 84.9		0.647
CA125 (U/mL) *	135.4 ± 375.7	192.5 ± 465.9	61.1 ± 204.6		0.100
Histology					0.017
Endometrioid carcinoma G1	47	19	28	40.4	
Endometrioid carcinoma G2	13	8	5	61.5	
Endometrioid carcinoma G3	9	8	1	88.9	
serous carcinoma	11	7	4	63.6	
carcinosarcoma	8	6	2	75.0	
sarcoma	3	3	0	100.0	
small cell carcinoma	1	1	0	100.0	
FIGO Stage					0.002
I	55	23	32	41.8	
II	1	1	1	100.0	
III	25	21	4	84.0	
IV	11	7	4	63.6	
Lympho-vascular invasion	40	30	10	75.0	0.002
Lymph node metastasis	21	14	7	66.7	0.281
Peritoneal cytology positive	23	15	8	65.2	0.328

CA19-9, carbohydrate antigen 19-9; CA125, carbohydrate antigen 125; FIGO, The International Federation of Gynecology and Obstetrics. * According to an analysis of variance (mean ± standard deviation).

## Data Availability

The data presented in this study are available upon request from the corresponding author. These data are not publicly available because of privacy restrictions.

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
