# Peer review of "Consistency between Primary Uterine Corpus Malignancies and Their Corresponding Patient-Derived Xenograft Models"

_ijms, 2024, doi:10.3390/ijms25031486_

Round 1
Reviewer 1 Report
Comments and Suggestions for Authors
Ueda et al. established PDX models for uterine corpus malignancies and analyzed the similarity between the primary and PDX tumors, through DNA and RNA sequencing analyses, and in terms of histological evaluation.
The results sound interesting, the method used is appropriate, the coverage of the material is good and the collected data have been discussed in an informative manner. The manuscript should be accepted for publication in the journal after minor revisions, on the following section:
- The authors should provide a graphical abstract.
- The keywords could be expanded up to a maximum of 10.
- In my best acknowledgment the dysregulated secretion and altered cargoes of EVs (i.e., proteins, lipids and nucleic acids) may be a sign of promoting tumor growth and metastasis. However, the knowledge about EV’s cargo subcellular/molecular sites of action in recipient cells is still fragmented. This is specifically true for proteins and nucleic acids associated with EVs. 1 So, the collected data by the authors are very interesting and have Scientific Soundness.
However, in the introduction, the readers may benefit from some more background and context about EV’s cargo, it would provide some orientation to readers and set the scene. Moreover, as has already been demonstrated, targeting EV biogenesis and secretion may have potential clinical implications for cancer metastatic therapy. For this purpose, I suggest seeing the following paper and adding the following references:
1. Int J Mol Sci. 2020 Jun 23;21(12):4463. doi: 10.3390/ijms21124463.
2. J Extracell Vesicles. 2021 Aug;10(10):e12132. doi: 10.1002/jev2.12132.
3. Nat Commun. 2023 Aug 10;14(1):4588. doi: 10.1038/s41467-023-40227-8.
- In vivo should be stated in italics.
Comments on the Quality of English LanguageMinor editing of English language required
Author Response
We appreciate the time and effort of the editor and referees in reviewing our manuscript. We have addressed all of the issues indicated in the review report and hope that the revised version meets the journal’s requirements for publication.
Response to Comments from Reviewer 1:
Comment 1:
The authors should provide a graphical abstract.
Response:
According to your suggestion, we created and submitted the graphical abstract.
Comment 2:
The keywords could be expanded up to a maximum of 10.
Response:
According to your suggestion, we added the keywords.
Comment 3:
In my best acknowledgment the dysregulated secretion and altered cargoes of EVs (i.e., proteins, lipids and nucleic acids) may be a sign of promoting tumor growth and metastasis. However, the knowledge about EV’s cargo subcellular/molecular sites of action in recipient cells is still fragmented. This is specifically true for proteins and nucleic acids associated with EVs. 1 So, the collected data by the authors are very interesting and have Scientific Soundness.
However, in the introduction, the readers may benefit from some more background and context about EV’s cargo, it would provide some orientation to readers and set the scene. Moreover, as has already been demonstrated, targeting EV biogenesis and secretion may have potential clinical implications for cancer metastatic therapy. For this purpose, I suggest seeing the following paper and adding the following references:
- Int J Mol Sci. 2020 Jun 23;21(12):4463. doi: 10.3390/ijms21124463.
- J Extracell Vesicles. 2021 Aug;10(10):e12132. doi: 10.1002/jev2.12132.
- Nat Commun. 2023 Aug 10;14(1):4588. doi: 10.1038/s41467-023-40227-8.
Response:
According to your suggestion, we added the sentences about EVs in introduction section. (page 2, line 68-73, 77-78)
Comment 4:
In vivo should be stated in italics.
Response:
We revised the words “in vivo” in italic. (page 2, line 57)
Reviewer 2 Report
Comments and Suggestions for Authors
The article by Shoko et al., titled "Consistency between Primary Uterine Corpus Malignancies and their Corresponding Patient-derived Xenograft Models," provides valuable insights into the potential platform for personalized medicine and precision medicine. However, several crucial points require clarification:
Major Issues:
* In Table 1, it's unclear how many biological repeats were conducted for each patient's tumor tissue in NOD/SCID mice. The authors should specify and define the criteria for the unsuccessful establishment of the PDX model, especially if more than one biological repeat was applied.
* Regarding Table 1, the engraftment rate does not show significance concerning the levels of CA19-9 and CA125. I recommend providing several ranges for CA19-9 and CA125 and correlating them with the engraftment rate. This would offer useful information for researchers aiming to establish these PDX models.
Minor Point:
* Move Figure 6 to the results section instead of the methods section.
Comments on the Quality of English LanguageN/A
Author Response
We appreciate the time and effort of the editor and referees in reviewing our manuscript. We have addressed all of the issues indicated in the review report and hope that the revised version meets the journal’s requirements for publication.
Response to Comments from Reviewer 2:
Comment 1:
In Table 1, it's unclear how many biological repeats were conducted for each patient's tumor tissue in NOD/SCID mice. The authors should specify and define the criteria for the unsuccessful establishment of the PDX model, especially if more than one biological repeat was applied.
Response:
The fragment resected from primary patient’ tumor was implanted in one mouse; we used only one mouse for PDX. We also used one mouse for implantation from PDX tumor to next generated PDX mouse; we did not repeat the implantation even if the tumor growth was not observed. According to your suggestion, we added the sentences about successful rate in method section. (page 13, line 494-497)
Comment 2:
Regarding Table 1, the engraftment rate does not show significance concerning the levels of CA19-9 and CA125. I recommend providing several ranges for CA19-9 and CA125 and correlating them with the engraftment rate. This would offer useful information for researchers aiming to establish these PDX models.
Response:
The engraftment rate was not significantly different between the patient with high level of CA125 or CA19-9 and those with low level of them. In our institution, the normal range was 37 U/mL or less for CA125 and 35 U/mL or less for CA19-9, respectively. When the patients with normal range of CA125 or 19-9 were considered as the low level group, the engraftment rate was not significantly different between the low and high level group (p=0.80 in CA125 and p=0.14 in CA19-9). The results were similar when the normal range was changed. According to your suggestion, the results were added. (page 2, line 85-87)
Comment 3:
Move Figure 6 to the results section instead of the methods section.
Response:
According to your suggestion, Figure 6 was moved to the result section.
Reviewer 3 Report
Comments and Suggestions for Authors
The manuscript titled "Consistency between Primary Uterine Corpus Malignancies and their Corresponding Patient-derived Xenograft Models" by Shoko Ueda et al. investigates the similarity between primary uterine corpus malignancies (UC) and their patient-derived xenograft (PDX) models. The study establishes PDX models from tissue fragments of patients with uterine corpus malignancies and performs comparative analyses using histological, immunohistochemical, DNA, and RNA sequencing methods. The results indicate that UC-PDX models retain the pathological, immunohistochemical, and genetic profiles of primary tumors, suggesting their potential as a platform for personalized medicine and translational research. This article discussed an issue in in cancer research by focusing on the development and analysis of PDX models for uterine corpus malignancies, a relatively under-explored area. The methods used for establishing the PDX models and the subsequent analyses (histological, genetic, etc.) are detailed and robust, contributing to the reliability of the findings.
The main issue with this paper is that, while clinical treatment of solid tumors is now entering the era of immunotherapy, the study is based on immunodeficient mouse models. This approach severely lacks consideration for tumor immunology-related issues, making it difficult to apply to the exploration of immunotherapy-related problems (such as ICIs). This leads to a disconnect between the research and the rapid development in clinical practice. The study has very low novelty and clinical significance, and therefore, it is not recommended for acceptance.
Other suggestions:
1.The manuscript should further discuss the limitations of the comparative analyses between primary tumors and PDX models, especially considering potential differences in tumor microenvironments.
2.Authors should further discuss the implications of these findings in the broader context of cancer research, treatment, and personalized medicine.
3.Consider including more comprehensive data on the response of these PDX models to various cancer treatments, which would enhance the practical applicability of the study in therapeutic contexts.
4.Provide more insights into potential future research directions, particularly in the application of these findings in clinical trials and drug development.

Author Response
We appreciate the time and effort of the editor and referees in reviewing our manuscript. We have addressed all of the issues indicated in the review report and hope that the revised version meets the journal’s requirements for publication.
Response to Comments from Reviewer 3:
Comment 1:
The main issue with this paper is that, while clinical treatment of solid tumors is now entering the era of immunotherapy, the study is based on immunodeficient mouse models. This approach severely lacks consideration for tumor immunology-related issues, making it difficult to apply to the exploration of immunotherapy-related problems (such as ICIs). This leads to a disconnect between the research and the rapid development in clinical practice. The study has very low novelty and clinical significance, and therefore, it is not recommended for acceptance.
Response:
The immunotherapy is widely used and discussed for cancer therapy as you mentioned. Humanized mouse may be useful for research of immunotherapy, however, humanized mouse is extremely expensive. Moreover, the span of life and research is very short; it is not useful of the PDX research for successful rate, pathological feature or gene analysis. There have been suitable mice for each kind of cancers, for example, the successful rate of gastric cancer PDX is similar between nude and other sever immune-deficient mice. SCID mice is usually used in prostate cancer. Then we chose immune-deficient mice for current study, however, humanized mice may be useful for research for immunotherapy. According to your suggestions, we added some sentences in limitation section. (page 12, 419-434)
Comment 2:
The manuscript should further discuss the limitations of the comparative analyses between primary tumors and PDX models, especially considering potential differences in tumor microenvironments.
Response:
According to your suggestion, we added the sentences in discussion section. (page 12, line 435-452)
Comment 3:
Authors should further discuss the implications of these findings in the broader context of cancer research, treatment, and personalized medicine.
Response:
According to your suggestion, we added the sentences in discussion section. (page 11, line 394-409)
Comment 4:
Consider including more comprehensive data on the response of these PDX models to various cancer treatments, which would enhance the practical applicability of the study in therapeutic contexts.
Response:
According to your suggestion, we added the comprehensive data on the response of these PDX models to various cancer treatments in discussion section. (page 11, line 386-394)
Comment 5:
Provide more insights into potential future research directions, particularly in the application of these findings in clinical trials and drug development.
Response:
According to your suggestion, we added our insights of future research direction in discussion section. (page 11, line 386-409)
Round 2
Reviewer 3 Report
Comments and Suggestions for Authors
Authors have improved and enriched the original content of the article based on the previous suggestions, making it more comprehensive. It is recommended for acceptance.
Author Response
We appreciate the time and effort of the editor and referees in reviewing our manuscript. We hope that the revised version meets the journal’s requirements for publication.